# LL-37-Coupled Porous Composite Scaffold for the Treatment of Infected Segmental Bone Defect

**DOI:** 10.3390/pharmaceutics15010088

**Published:** 2022-12-27

**Authors:** Xialin Li, Xingyu Huang, Long Li, Jiayi Wu, Weihong Yi, Yuxiao Lai, Lei Qin

**Affiliations:** 1Department of Orthopedics, Huazhong University of Science and Technology Union Shenzhen Hospital, Shenzhen 518052, China; 2Centre for Translational Medicine Research & Development, Shenzhen Institutes of Advanced Technology, Chinese Academy of Sciences, Shenzhen 518055, China; 3Shenzhen Institutes of Advanced Technology, University of Chinese Academy of Sciences, Shenzhen 518055, China; 4Department of Emergency Department, Huazhong University of Science and Technology Union Shenzhen Hospital, Shenzhen 518052, China

**Keywords:** peptide LL-37, non-antibiotic antimicrobial graft, PLGA/TCP/LL-37 scaffolds, low-temperature 3D-printing technology, rat infected femoral defect model

## Abstract

Increased multiantibiotic-resistant bacteria means that infected bone defects remain a significant challenge to clinics. Great interest has emerged in the use of non-antibiotic antimicrobials to reduce the rate of multiantibiotic-resistant bacterial infection and facilitate bone regeneration. The cationic antimicrobial peptide LL-37 is the sole human cathelicidin and has shown nonspecific activity against a broad spectrum of microorganisms. In this study, we fabricated the poly(lactic-co-glycolic acid)/β-calcium phosphate/peptide LL-37 (PLGA/TCP/LL-37, PTL) scaffold with low-temperature 3D-printing technology for the treatment of infected segmental bone defects. The prepared scaffolds were divided into three groups: a high LL-37 concentration group (PTHL), low LL-37 concentration group (PTLL) and blank control group (PT). The cytocompatibility and antimicrobial activity of the engineered scaffolds were tested in vitro, and their osteogenesis properties were assessed in vivo in a rat infected bone defect model. We found the fabricated PTL scaffold had a well-designed porous structure that could support a steady and prolonged LL-37 release. Furthermore, the PTHL group showed strong antibacterial activity against *Staphylococcus aureus* (*S. aureus*) and *Escherichia coli* (*E. coli*) without any inhibition of the proliferation or alkaline phosphatase activity of rat bone marrow mesenchymal stem cells (BMSCs) in vitro. In addition, the infected femoral defects implanted with PTHL group displayed new bone formation in four weeks without any evidence of residual bacteria, which showed similar antibacterial outcomes to the vancomycin and cancellous bone mixture group. In conclusion, the PTHL composite scaffold is a promising non-antibiotic antimicrobial graft with good biodegradability, biocompatibility, and osteogenic capability for infected bone defects.

## 1. Introduction

Autologous, allograft and artificial bone grafts are the three main kinds of bone grafts used in clinics. Autografts are still considered the gold standard for bone substitution, but their disadvantages such as limited supply, donor site pain and infection limit its application [1,2]. Allografts face the potential risk of local immune reaction and infection, which ultimately leads to bone nonunion and surgical failure [1,3,4]. Poly(lactic acid-co-glycolic acid) (PLGA) is a high-molecular copolymer approved by the FDA for its usage in human implants [5], which has been widely used in drug delivery, tissue engineering, and medical and surgical equipment for its good biocompatibility, biodegradability and non-toxicity properties [6]. Beta-tricalcium phosphate (β-TCP) is known for its excellent osteoconductivity, biocompatibility and low toxicity [7,8]. Moreover, the chemical structure of β-TCP is similar to that of hydroxyapatite (HA), which is the intrinsic inorganic substance in bone tissue [9]. Therefore, the biological bone scaffold composited with both of β-TCP and PLGA has been demonstrated to have a beneficial bionic bone effect in repairing bone defects [10,11].

Infection is one of the key causes of nonunion in large bone defects after orthopedic surgery. According to the published data, the overall rate of infection in implants is about 2–5% [12,13]. Moreover, this number could reach up to 30–50% in open fractures [14]. Current treatments for the postoperative infection of a fractured bone include debridement, implant removal, systemic antibiotic therapy and regraft to fill the aseptic bone defect [15,16]. However, these traditional treatments can lead to the second surgical trauma of patients, high medical costs and the consumption of medical resources. Unfortunately, some patients have to endure more than two operations, and repeated antibiotic use is likely to induce local drug-resistant strains that ultimately lead to treatment failure [14,15]. Since composite biological bone is difficult to endow with antimicrobial properties, the integration of antimicrobial strategies is widely used to combat bone infections.

Antibiotics are currently the most effective antibacterial strategy for the treatment of bone-implant-associated infection. However, the systemic administration of antibiotics in clinics leads to drug-resistant bacteria and a limited ability to eradicate challenging biofilm-forming pathogens, such as *Staphylococcus aureus* (*S. aureus*) as the most common cause of osteomyelitis [15,17]. Since systemic administration can cause serious side effects and bacterial resistance, potential antibacterial properties can be integrated into bone grafts with local antibacterial strategies to eliminate the risk of postoperative infection. Currently, a variety of antibacterial bone grafts have been developed based on different antibacterial mechanisms, including composite biological bone scaffolds with drug-induced antibacterial functions, composite biological bone scaffolds with ion-mediated antibacterial functions, and composite biological bone scaffolds with physical antibacterial functions.

Although drug-induced antimicrobial strategies are still the most effective and widely used approach in clinical applications, bacterial resistance remains a major challenge. For example, an ion-mediated antibacterial strategy can achieve a good antibacterial effect due to its ability to continuously release antibacterial ions, such as copper [18] and magnesium [19]. As the ions used in this strategy are mostly heavy metals, it faces biosafety risks and great obstacles to clinical application [17]. Moreover, the antibacterial mechanisms of a physically activated antibacterial strategy are still not clear [17]. Therefore, the development of antimicrobial alternatives has become particularly important.

The antimicrobial peptide (AMP), a new generation of antibiotics produced by species across the tree of life, has gained popularity in recent years in the therapeutics [20,21]. For example, human β-defensin 3 can significantly reduce bacterial load, and infections in rats treated with β-defensin 3 were cleared to the same extent as those treated with the antibiotic vancomycin [22,23]. Moreover, there are four common structures of AMPs, including helix-based, sheet-based, coil-based and composite, whose modifications are used to identify different types of AMPs [24]. The molecular mechanisms of the antimicrobial functions of AMPs include cell wall action, membrane-targeting action, transmembrane mechanism of intracellular action and intracellular action targets [25]. In addition, besides their antibacterial activities, AMPs have great performance in terms of antibiofilm activity, immunomodulation and regenerative properties [24].

LL-37 is the only naturally occurring antimicrobial peptide in the human body, which is composed of 37 amino acid residues linked together [26]. LL-37 is cleaved from the C-terminal end of the human cationic antimicrobial protein 18 (hCAP-18), and widely found in monocytes and neutrophils of the bone marrow and blood, as well as the epithelium, skin, mucous membrane, and testes [27,28]. Published results have shown that LL-37 has a broad spectrum of antibacterial activity, immunomodulatory activity and osteogenic induction ability [29,30]. Studies have revealed that LL-37 has antibacterial activity that exhibits a broad spectrum of bactericidal activity against both Gram-positive and Gram-negative bacteria [31,32]. Meanwhile, LL-37 has potent chemotactic and immunomodulatory properties. Cathrine et al. reported that tissue- and host-specific variations in LL-37 concentrations may influence vancomycin susceptibility in vivo [33]. Yu et al. reported that LL-37 promoted the proliferation, migration and osteogenic differentiation of bone marrow stromal cells (BMSCs), and attenuated the osteogenic inhibition induced by lipopolysaccharide (LPS) [32]. In addition, LL-37 has a variety of biological functions, including angiogenesis, the promotion of wound healing and the inhibition of osteoclastogenesis [34].

In the current study, we used low-temperature 3D-printing technology to manufacture PLGA/β-TCP containing different concentrations of LL-37, and evaluated its physical properties, biocompatibility, osteogenic induction, and sustained-release ability of active peptides. We also evaluated its ability to treat infectious bone defects in a rat femur infectious defect model. Our results showed that the low-temperature 3D-printing technology not only displayed porous architectures but also substantially maintained the bio-efficacy of LL-37. We also demonstrated that infected femoral defects implanted with PTHL group displayed new bone formation in four weeks without any evidence of residual bacteria.

## 2. Materials and Methods

### 2.1. Chemicals and Reagents

PLGA (lactide-to-glycolide ratio: 75:25) was provided by Chongqing Yusi Pharmaceutical Technology (Chongqing, China); β-TCP was from Shanghai Aladdin Biochemical Technology (Shanghai, China); LL-37 was purchased from Eurogentec; a-minimal essential medium (a-MEM) and fetal bovine serum (FBS) were purchased from Gibco (Waltham, MA, USA) for cell culture; LL-37 ELISA kit was obtained from Sigma-Aldrich (St. Louis, MO, USA); CCK-8 kit was obtained from Shanghai Bebo Biological Technology (Shanghai, China); ALP staining kit was purchased from Shanghai Biyuntian Biotechnology (Shanghai, China).

### 2.2. Preparation of Porous PTL Composite Scaffolds

A low-temperature 3D printer was provided by Hangzhou Jinuofei Biotechnology Co., Ltd., Hangzhou, China. We used the computer-aided artificial bone scaffold design system to set the basic parameters of the required scaffold. Then, the preset file of the scaffolds was imported into the low-temperature 3D-printing system, and parameters such as the moving speed of the nozzle and the printing layer thickness were set during the printing process. The compound material of PLGA/β-TCP/LL-37 was loaded into the printing cartridge to prepare the composite scaffolds. The prepared scaffolds were divided into three groups: PLGA/β-TCP/high concentration LL-37 group (PTHL), PLGA/β-TCP/low concentration LL-37 group (PTLL) and PLGA/β-TCP blank group (PT). PLGA was dissolved in 10 mL 1,4-dioxane (relative density is 1.03), then ultrasonically dissolved in an ice-water bath for 30 min followed by magnetic stirring for 4 h. β-TCP was added and stirred for 2 h until mixed well. Then, 1 mL LL-37 was added into the configured PLGA/β-TCP print ink and stirred with magnetic force for 2 h. The mass ratio of PLGA:TCP:LL-37 was 160:40:0.11 in the PTHL group, 160:40:0.022 in the PTLL group and 160:40:0 in the PT group.

### 2.3. The Surface Morphology of Scaffold

The morphology and pore sizes of the scaffold were observed by scanning electron microscopy (SEM, Hitachi SU8010). Briefly, we prepared the PTL composite scaffolds and let them air-dry overnight. The next day, the scaffolds were mounted using double-sided conductive tape, and coated with Au/Pt (Gold/Platinum) particles in a pumper to increase the sample conductivity. The surface morphology of scaffolds was observed on a scanning electron microscope at 5.0 kV.

### 2.4. Degradation Assay In Vitro

The scaffolds of each group were 3D-printed and washed with PBS. The weight of each scaffold was measured before being soaked in 10 mL PBS solution. The scaffolds emerged in PBS were placed in a 37 °C shaker. The scaffolds were weighted every 2 days for 22 days. The ratio of weight loss was calculated as follows:Weight loss (%) = (M_0_ − M_n_)/M_0_ × 100%

### 2.5. Cytotoxicity, Osteo-Promotive Effects and LL-37 Release In Vitro

#### 2.5.1. CCK-8 Cell Proliferation Assay

Pre-osteoblastic MC3T3-E1 cells (subclone 4, ATCC CRL-2593) were purchased and maintained in a-minimal essential medium (a-MEM) and 10% fetal bovine serum (FBS) at 37 °C with 5% CO_2_. Scaffolds of each group were cut into 3 × 3 × 3 mm^3^ samples and sterilized with UV for 30 min on each side, then 2 × 10^4^ cells were seeded into each sample for testing. Each group had 5 replicates. Following the specifications, after incubation in growth medium at 37 °C for 72 h, the culture medium of the samples was replaced with 100 mL medium (content 10% CCK-8) and incubated at 37 °C for 1 h. Then, the microplate analyzer was adjusted to 450 nm to measure the OD value of each well.

#### 2.5.2. ALP Active Assay

Pre-osteoblastic MC3T3-E1 cells were seeded onto scaffolds of each group and cultured in osteogenic induction medium (DEME/f12 + 10% FBS + 1% double antibody + 100 nM dexamethasone + 50 μM vitamin C + 10 mM β-disodium glycerophosphate) for 14 days. Normal saline was used in the control group. Each group had three replicates. Then, the culture medium was discarded and washed 3–5 times. ALP staining was conducted following the manuscript protocol (Cat# CTCC-JD002, Wuxi Puhe Biotechnology, Wuxi, China).

#### 2.5.3. LL-37 Release In Vitro

The release of LL-37 from the PTHL or PTLL scaffold was tested in vitro in PBS (pH 7.4) at 37 °C. Initially, 10 mg biomaterial was immersed in 1 mL PBS in a 1.5 mL EP tube. At each time point (1, 3, 6, 9, 12, 18, and 21 days), the tubes were centrifuged at 800 rpm for 30 min and the supernatant was collected and replaced by 1 mL of fresh PBS. Human LL-37 ELISA kit was used to detect the concentration of LL-37 in PBS solution.

### 2.6. Antimicrobial Activity In Vitro

A Kirby–Bauer Test was used to test the ability of various scaffolds to inhibit microbial growth. Briefly, the *E. coli* (group A) and *S. aureus* (group B) were diluted in PBS to 5.0 × 10^6^ CFU/mL. A single bacterium strain was spread over an agar plate using a sterile swab, then incubated in the presence of the antimicrobial object. Gentamicin tablets were used as the positive control in group A, vancomycin tablets were used as the positive control in group B, and normal saline filter paper was used as the negative control. All of the scaffolds were cut into small rounds 5 mm in diameter and sterilized. After being cultured at 37 °C for 16 h, the diameters of various bacteriostatic zones were measured. The antibacterial results of the experimental group were determined according to WS/T 650-2019 standard: a diameter of bacteriostatic zone more than 7 mm was judged to have antibacterial effect, and less than 7 mm was judged to have no bacteriostatic effect.

### 2.7. Osteo-Promotive Effects of the PTL Scaffolds In Vivo

Different groups of scaffolds were implanted to repair large, infectious bone defects in rat femurs. Specifically, 24 Sprague Dawley (SD) rats (6 weeks old, male) were randomly divided into four groups. One experienced surgeon performed all the operations in aseptic conditions. Under general anesthesia by intraperitoneal injection of Ketamine (80 mg/kg) and xylazine (10 mg/kg), a 1 cm longitudinal incision in the right lateral femur was chosen. After separating the muscle tissues, the lateral femoral condyle was exposed. Referring to the previous method, a 4 mm × 4 mm unicortical hole was generated at the condyle with a dental bur, then 10^8^ CFU *S. aureus* in 75 μL 20% gelatin gel and 50 mg biomaterials were implanted into the bone defect. The overlying fascia were sutured to close defect and then the muscle and skin were stitched. After the operation, rats were independently housed in cages and allowed to eat and drink. Enrofloxacin (10 mg/kg) was administered for 3 days to prevent infection. One month later, bone samples were harvested. All animal experimental protocols complied with all relevant ethical regulations and approved by the Animal Care and Committee of Shenzhen Institutes of Advanced Technology (No: SIAT-IACUC-191125-YGS-ZW-A0949).

#### 2.7.1. Micro-CT Analysis of Trabecular Architecture

One month after the operation, the rats were euthanized according to regulations, and the femur was removed and soaked in 4% paraformaldehyde solution for 3 days. It was then transferred to phosphate buffer solution for cleaning and placed in a high-precision micro-CT instrument (Skyscan1172, Kontich, Belgium) as required. Micro-CT scan parameters were set to 65 kV voltage, 153 μA current, 6.65 μm accuracy. Image reconstruction and analysis were performed using NRecon V1.6, CTAn V1.9 and CTVol V2.3 software (Skyscan, Belgium). All samples were selected for the analysis of the original bone defect at the lower end of femur, and finally the parameters of bone volume/tissue volume (BV/TV) and bone trabecular thickness (Tb.Th) were selected for statistical analysis.

#### 2.7.2. Histological Evaluation

IHC experiments were performed following our standard protocol as described in [35]. The femurs were collected and fixed in 4% paraformaldehyde overnight and decalcified with 10% EDTA (pH, 7.4) for 21 days. The samples were then dehydrated with 30% sucrose for 24 h and embedded in paraffin. We prepared 4 μm-thick coronal-oriented sections for immuno-histochemistry staining using a standard protocol. Briefly, the sections were deparaffinized and rehydrated by sequentially immersing them in 100% Xylene, 100% ethanol, 95% ethanol, 75% ethanol, and deionized H_2_O solution (5 min for each). Then, the sections underwent the antigen-retrieval procedure by immersing them in sodium citrate buffer at 58 °C overnight. The next day, the sections were cooled to room temperature and washed with deionized H_2_O twice. The sections were covered with peroxidase (Dako, Hovedstaden, Denmark) for 20–30 min in a humid dark box, and washed with PBS twice. The sections were blocked with prediluted normal horse serum for 1 h, then incubated with the primary antibodies for mouse osteocalcin (Takara Bio, M173, 1:200) in a humid dark box overnight at 4 °C. The next day, the sections were washed in PBS three times, 5 min each. Then, the sections were incubated in prediluted biotinylated secondary antibody (Vector Lab, CA, USA) in a humid dark box at RT for 30 min, followed by washing for 5 min in PBS. Then, the sections were incubated in DAB peroxidase substrate (Vector Lab, CA, USA) solution until the desired stain intensity developed, followed by counter-staining with hematoxylin (Dako, Hovedstaden, Denmark). We used an Olympus BX51 microscope for sample image capturing. Image J software was used to analyze the IHC image.

### 2.8. Statistical Analysis

SPSS18.0 software was used for statistical analysis of the obtained experimental data. The significance and statistical significance of the difference between the two groups of data were evaluated by unpaired, two-tailed Student’s *t*-tests. The data were expressed as mean ± standard deviation (SD), and *p* < 0.05 indicated that the differences were statistically significant.

## 3. Results

### 3.1. Morphology of the PTL Scaffolds

We successfully prepared PTL antibacterial artificial bone scaffolds through low-temperature 3D printing (Figure 1A,B). The rough scaffold surface was observed under a scanning electron microscope (Figure 1C), which was beneficial for the implantation and attachment of osteoblasts. No granular or powdery structures were found on the surface of the scaffolds. All the scaffolds had regular porous structures with interconnected pores with a relatively equal pore size about 300 μm.

### 3.2. Biocompatibility, Degradation Rate, Osteo-Inductivity and Controlled Release of the PTL Scaffolds In Vitro

MC3T3-E1 cells were co-cultured with scaffolds in maintenance medium. After three days, the proliferation ability was detected by the CCK-8 method. There was no significant difference among the PT, PTLL, PTHL and blank control groups (Figure 2A). To verify the osteo-promotive effects, the ALP activity was determined in order to evaluate the potential osteogenic ability of MC3T3-E1 on those scaffolds. After cultured in osteogenic induction medium for 14 days, the scaffold group showed similar ALP activity compared with the control group (Figure 2C). These results suggest that low or high LL-37 concentrations do not influence the osteogenic differentiation of the MC3T3-E1 cells on scaffolds.

The degradation rate of each scaffold in PBS solution is shown in Figure 2B. Within the first 10 days, the weight of the scaffolds decreased slowly to more than 90%. After that, their degradation velocity began to accelerate and the rate of weight loss reached about 35% at three weeks, and there was no significant difference between the PT, PTLL and PTHL groups.

To investigate the duration of LL-37 sustained release from the PTHL and PTLL scaffolds, the cumulative release rate was measured (Figure 2D). Within three weeks, LL-37 was released slowly at a relatively uniform rate, and the cumulative release rate reached about 50% at the end of three weeks. The PTHL and PTLL groups showed similar sustained release rates.

### 3.3. Antimicrobial Activity In Vitro

We observed the antibacterial activity of PTL scaffolds through the Kirby–Bauer test. In the *E. coli* AGAR plate, the PTHL scaffold showed a medium-sized bacteriostatic circle, of which the diameter was less than that of the gentamicin tablet, and there was no bacteriostatic circle in the PT or PTLL scaffolds (Figure 3A). Similarly, in the *S. aureus* AGAR plate, the PTHL scaffold also showed an obvious bacteriostatic circle, of which the size was similar to that of the vancomycin tablet (Figure 3B). At the same time, a small-sized circle emerged in the PTLL scaffold. These results verified that the PTHL scaffold had antimicrobial activity to both *E. coli* and *S. aureus* strains in vitro.

### 3.4. Osteo-Promotive Effects of the PTL Scaffolds In Vivo

#### 3.4.1. Radiography

The rats were randomly divided into five groups, which included the model group (model of infectious bone defect without any material implantation), PT group (infective bone defect model was implanted with the PT scaffold), PTLL group (the model of infectious bone defect was implanted with the PTLL scaffold), PTHL group (the model of infectious bone defect was implanted with the PTHL scaffold) and positive control group (cancellous bone granules mixed with freeze-dried vancomycin powder were implanted into the infected bone defect femur).

As shown in Figure 4A, there was no bone formation in the infected bone defect in the model group according to micro-CT analysis. For the PT and PTLL groups, no new bone was found in the infected bone defect, and no absorption of the filling material was found in the bone defect. There was marked osteosclerosis around these scaffolds. Comparably, the PTHL group showed good antibacterial activity and osteogenic induction after one month of implantation. There was obvious bone growth in the bone defect, and the filling material was partially absorbed and replaced by new bone crawling. There was no obvious osteosclerosis around the residual scaffold. In the positive control group, there was obvious new bone growth in the bone defect. By comparison, the PTHL group and positive control group displayed significantly newly formed bone compared with the other groups, indicating that these two bone grafts controlled infection and induced new bone growth. A quantitative analysis of micro-CT showed that the PTHL scaffold could enhance more new bone formation (BV/TV, Tb.Th) compared with the PT and PTLL scaffolds and the model group (Figure 4B,C).

#### 3.4.2. Histological Analysis

Osteocalcin (OCN) staining was performed to evaluate new bone quality in the various groups four weeks after the operation (Figure 5). In the model group, fibrous tissue only partially generated around the large bone defect and no new bone tissue emerged in the defect. In the PT group, the scaffold in the defect was not absorbed at all and many bacteria remained on the surface of the material. Similarly, no new bone formation was observed. In the PTLL group, the material was encased in fibrous tissue but fewer bacteria were observed. It seemed that part of the material was absorbed but no new bone tissue replaced it. In the PTHL group, most of implant material was absorbed and obvious new bone tissue growth crawling replacement was seen, without bacteria or fibrous scar tissue in the defect. In the positive control group, bone defects were locally filled with new-formed bone, which was similar to the PTHL group. In short summary, these results indicate that the PTHL scaffold and positive control material have better antibacterial and osteogenic effects than the other groups, which is consistent with the radiography analysis results.

## 4. Discussion

In the current study, we designed a composite PTL scaffold with potential antibacterial activity using low-temperature 3D-printing technology, which can continuously release LL-37. The microstructure of the PTL scaffold is similar to that of natural human cancellous bone, and its biocompatibility and degradation rate are suitable for the inward growth and survival of cells. Furthermore, in the infected bone defect rat model, we found that the PTHL scaffolds could support new bone formation and inhibit bacterial growth through micro-CT analysis and OCN IHC staining experiments. These results suggest that the PTL scaffold is a promising implant with antibacterial properties that can inhibit bacterial growth and promote osteogenesis in vivo.

To our knowledge, our study could be the first report regarding the combination usage of LL-37 and a PLGA/β-TCP scaffold for the treatment of infectious bone defects. Due to the efficient antibacterial performance and multi-functional ability of LL-37, this natural human AMP has gained much attention in its application for infected bone defects. Recently, published data showed that the injection of LL-37 into rabbits helped the internal femoral fracture fixation upon infection [36]. Titania nanotubes (NTs) loaded with LL-37 significantly promoted osteogenesis in both uninfected and infected rat models [37]. Moreover, a combination scaffold of Poly(sebacoyl diglyceride) (PseD), human adipose-derived mesenchymal stem cells (hADSCs) and LL-37 significantly accelerated the process of bone reconstruction in a rat calvarial bone defect model [38]. In the current study, low-temperature 3D-printing technology was used to prepare a PLGA/β-TCP composite scaffold, and the antibacterial peptide LL-37 was loaded to prepare an artificial antibacterial bone graft.

Beta-tricalcium phosphate (β-TCP) is known for its chemical structure similar to that of hydroxyapatite (HA) and its osteoconductivity. The faster in vivo degradation properties of β-TCP make it more suitable for biodegradable biomaterials than HA. The incorporation of PLGA into this polymer was aimed at improving its bioactivity and osteoconductivity, which has been demonstrated in other studies. Besides, β-TCP can prevent the patient’s physiological environment from becoming acidic due to the degradation of PLGA [9,10]. In this study, the PLGA/β-TCP scaffolds were prepared by combining PLGA and β-TCP with a mass ratio of 4:1 using 3D-printing technology. PLGA/β-TCP scaffolds with an aperture size of 300 μm, a pore size of the biomaterial scaffolds larger than 300 mm, and a porosity higher than 50% are preferred as bone substitute implants because of their good vascularization, which results in direct reparative osteogenesis [7]. Both in in vitro and in vivo experiments of the current study, the scaffolds showed good biocompatibility, osteoconductivity and biodegradation. Moreover, the infected rat femoral defect models showed that the PTHL group displayed excellent antimicrobial activity and osteo-inductive capability one month after the bone defect operation. There was obvious bone in-growth in the femur defect of the experimental rats, and the filling scaffold was partially absorbed and substituted by newly formed bones. No obvious osteoproliferation nor osteosclerosis was found around the residual scaffold.

Future studies based on the current research could apply a longer observation period with better in vivo performance to test mixed bacterial species infection, and to explore the underlying detailed cellular and molecular mechanisms. In this study, the rat model was only monitored for one month after surgery, and neither bone in-growth nor full reconstruction was observed. It would be great to extend the experimental time to two or three months for a longer monitor period. Moreover, a 3D-bioprinted drug-delivery scaffold has been widely accepted as a promising biomaterial in bone tissue engineering [39]. It is encouraged to optimize the best ratio between PLGA and β-TCP, as well as to optimize the LL-37 concentration loaded on the scaffold in the treatment of infected bone defects. Furthermore, in clinics, chronic osteomyelitis in patients is often a mixed bacterial infection. We only used *S. aureus* as the infected bacteria in the current study, so a future study is suggested to test the performance of the PLGA/β-TCP/LL-37 scaffold in mixed-infection situations. In addition, our IHC results showed significant osteogenesis detected in the PTHL group, which suggested a higher new-bone-formation rate occurring in and around the implanted PTHL scaffold. It would be interesting to further examine whether PLGA/β-TCP/LL-37 stimulates resident BMSC or osteoblast proliferation, differentiation, or in-growth into scaffolds, and whether this contributes to overall bone fracture healing.

## 5. Conclusions

In this study, we introduced a new combined scaffold of PLGA/β-TCP with the antibacterial peptide LL-37. This scaffold has a porous morphology, is safe for osteoblast cell growth and function, and its slowly released LL-37 contributed to antibacterial function in both in vitro and in vivo experiments. Our study demonstrated that the PLGA/β-TCP/LL-37 composite scaffold is a new promising non-antibiotic antimicrobial graft with good biodegradability, biocompatibility, and osteogenic capability for infected bone defects, which is expected to be applied to clinical practice as a local bone-filling material in the near future.

## Figures and Tables

**Figure 1 pharmaceutics-15-00088-f001:**
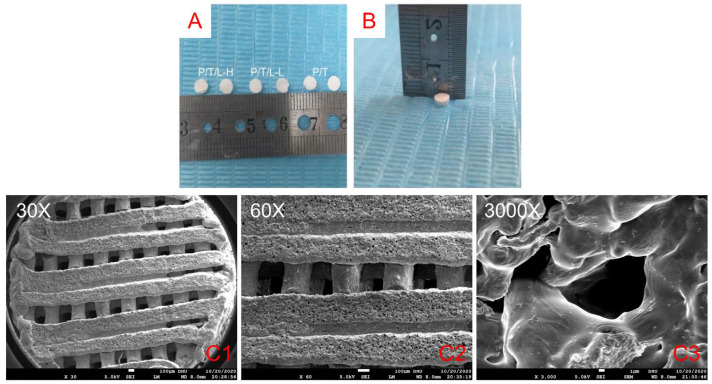
Morphology of the PTL scaffolds. Artificial bone scaffolds prepared by low-temperature 3D printing. The diameters of PTHL, PTLL and PT scaffolds (**A**); the scaffold height of the blank control group (**B**). The PTHL scaffold was magnified 30 times (**C1**); 60 times (**C2**); 3000 times (**C3**) under a scanning electron microscope.

**Figure 2 pharmaceutics-15-00088-f002:**
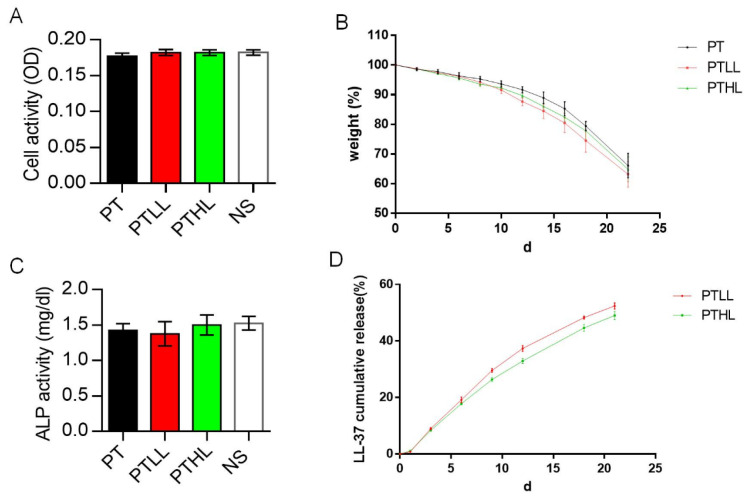
Biocompatibility, degradation rate, osteo-inductivity and controlled release of the PTL scaffold in vitro. (**A**) The cell proliferation of MC3T3-E1 cells on scaffolds measured by CCK-8 kit (*n* = 5). (**B**) Degradation rate of PT, PTLL and PTHL scaffolds (*n* = 3). (**C**) ALP activity of MC3T3-E1 cells in PT, PTHL PTLL scaffolds and control group after induction for 14 days (*n* = 3). (**D**) Accumulated release of LL-37 in vitro (*n* = 3).

**Figure 3 pharmaceutics-15-00088-f003:**
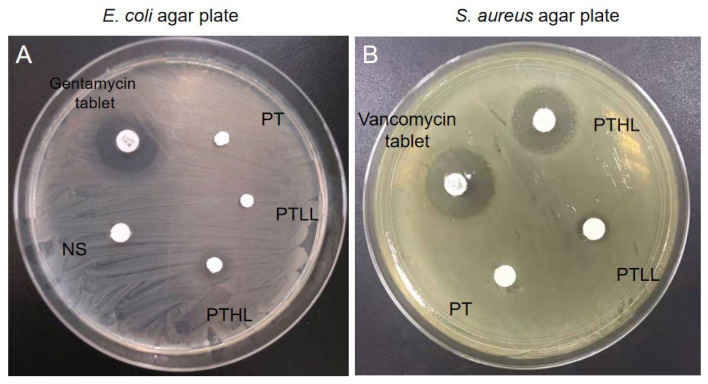
In vitro antibacterial activity of PLGA/TCP/LL-37 grafts in *E. coli* (**A**) and *S. aureus* (**B**) agar plates. Gentamycin tablets and vancomycin tablets were used as positive controls; PT is the PLGA/β-TCP blank group; PTLL is the PLGA/β-TCP/low concentration LL-37 group; PTHL is the PLGA/β-TCP/high concentration LL-37 group; NS is the tablet with normal saline group, as the negative control in this experiment.

**Figure 4 pharmaceutics-15-00088-f004:**
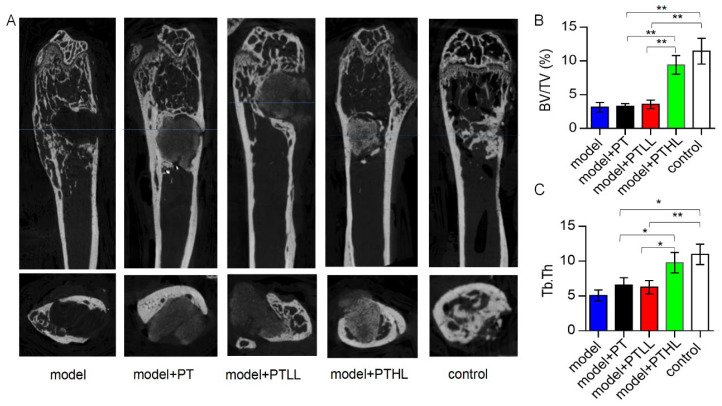
Micro-CT of *S. aureus*-infected mice femoral segmental defects implanted with various scaffolds. (**A**) Representative coronal plane and horizontal plane images of each group. Comparison of BV/TV (**B**) and Tb.Th (**C**) of each group. The parameters for new bone formation included the bone volume fraction (BV/TV, %, as the volume of mineralized bone per unit volume of the sample) and trabecular thickness (Tb.Th, mm, as the thickness measurement of cancellous bones). Student’s *t*-test, mean ± sd, * *p* < 0.05, ** *p* < 0.01.

**Figure 5 pharmaceutics-15-00088-f005:**
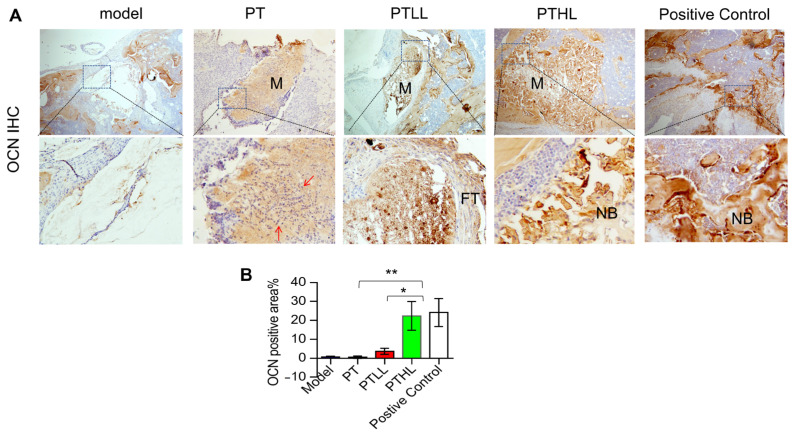
OCN expression in rat femoral defect models. (**A**) Histological images of OCN IHC staining in infected femoral defects (red arrow: bacteria; M: implanted material; FT: fibrous tissue; NB: new bone tissue). (**B**) The percentage of OCN-positive area in IHC image of each group. Student’s *t*-test, mean ± sd, * *p* < 0.05, ** *p* < 0.01.

## Data Availability

All data generated or analyzed during this study are included in this article. Any other data are available from the corresponding author upon reasonable request.

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
