# Peer review of "LL-37-Coupled Porous Composite Scaffold for the Treatment of Infected Segmental Bone Defect"

_pharmaceutics, 2022, doi:10.3390/pharmaceutics15010088_

Round 1

Reviewer 1 Report

The manuscript entitled “LL-37 coupled porous composite scaffold for the treatment of infected segmental bone defect” is very interesting and in a field that is rapidly developing. The submitted manuscript describes some interesting results giving the solution for bacteria infections after scaffold implantation without the need for the usage of antibiotics. However, in this form submitted manuscript is not ready for publication. The abstract is very well written. Only the names of bacteria should be written in italic. This should be applied further in the manuscript also.
In the introduction (at the and) the information regarding the novelty of this study is missing. Is this the first time that this type of research was conducted? Why these types of scaffolds were used?
Was this scaffold previously prepared and published in the literature? If not, more physiochemical characterization of the scaffold is required.
Materials and methods do not provide enough information related to materials preparation and characterization. For example, there is no information on how the scaffold was coated before SEM analysis. What were conditions used during analysis, etc. Further, not enough information was given for degradation analysis. Protocols for biological characterization also need to be improved. Authors need to have in mind that according to the Materials and method section other researchers need to be able to repeat the reported (submitted) study.
Overall, the description of the discussion needs to be expanded. In addition, the discussion is too short for this type of study. In other words, the discussion should be expanded and strengthened if authors would like that study is publishable in a journal Pharmaceutics with a high IF.
What are the drawbacks of this study?

Reviewer 2 Report

The authors present a clear and comprehensive study of bone scaffolds impregnated with antimicrobials.  The following minor issues should be addressed:

1. The antimicrobial peptide field is very diverse with sequences.  The authors should expand the introduction to discuss more of the mechanism of action of peptides, and how using peptides of different size or compositions may help functionality

2. The authors only cite one review article in the introduction regarding ion-releasing treatments.  I recommend adding: (https://doi.org/10.1111/jam.13681 ; https://doi.org/10.3390/polym13132183;  https://doi.org/10.3390/molecules22091487  )

3. The figure legend for Figure 3 is confusing.  The wording should be adjusted to clarify the difference between (A) and (a), etc.  as well as delineating these differences

4. There are several examples of missing italics on bacterial names.

5. In figure 4, the authors should describe the axis abbreviations for BV/TV, Tb.Th in the legend itself.

6. Overall there should be a review of the text for tense/pluralization issues and article usage.

Round 2

Reviewer 1 Report

-